# Hormonal and Neurological Aspects of Dog Walking for Dog Owners and Pet Dogs

**DOI:** 10.3390/ani11092732

**Published:** 2021-09-18

**Authors:** Junko Akiyama, Mitsuaki Ohta

**Affiliations:** 1Department of Animal Health Technology, Yamazaki University of Animal Health Technology, Tokyo 192-0364, Japan; j_akiyama@yamazaki.ac.jp; 2School of Veterinary Medicine, Azabu University, Sagamihara-shi 252-5201, Japan

**Keywords:** walk, dog owners, pet dogs, saliva, oxytocin, MHPG, GABA

## Abstract

**Simple Summary:**

Introduction: Dog walking is a common activity for dog owners and their dogs. We investigated the effects of dog walking on both owners and dogs, focusing on salivary oxytocin and cortisol (Experiment I) and brain neural activity (Experiment II). Methods: We collected saliva samples from 34 pairs of pet dogs and their owners at four time points (before dog walking, 15 min into the walk, at the end of the 30 min walk, and 10 min after the end of the walk). As a control, we assessed dog owners who took a walk without their dogs. We measured salivary oxytocin and cortisol with enzyme-linked immunosorbent assay kits, monoamines, and GABA with high performance liquid chromatography. Results: In Experiment I, walking with a dog, relative to walking without a dog, did not affect the owners’ salivary oxytocin and cortisol, as in previous research. However, in Experiment II, walking with a dog decreased the owners’ salivary MHPG compared to walking without a dog. MHPG correlated negatively with GABAergic activity. Discussion and Conclusions: Dog walking did not boost the owners’ salivary oxytocin or cortisol but did inhibit brain noradrenergic nerves via GABA activity, suggesting stress relief.

**Abstract:**

The hormone oxytocin is involved in various aspects of the relationship between humans and animals. Dog walking is a common activity for dog owners and their dogs. The walk, of course, should be good for the health of the dog as well as its owner. In Experiment I, we assessed whether salivary oxytocin and cortisol in dog owners changed because of walking their dogs. Ten owners walked with their dogs and walked alone. Similar to other previous research, walking with a dog did not significantly change oxytocin and cortisol. Therefore, in Experiment II, we investigated the effect of dog walking on brain noradrenergic and GABAergic neural activity, as indicated by salivary MHPG and GABA, in 14 dog owners. Walking with a dog reduced salivary MHPG compared to walking alone, and MHPG was correlated negatively with GABA. Thus, dog walking activated GABAergic nerves in the brain and suppressed noradrenergic nerves, effectively relieving stress.

## 1. Introduction

Dogs and other companion animals have healing effects on people, such as producing enjoyment and stress relief [1,2,3]. Considering that humans have lived with dogs longer than any other domesticated animals [4], the healing effects of human–dog relationships have been taken for granted. However, scientific research recently has begun to address the matter [5].

Researchers have posited that the intimate relationship between owners and their dogs is similar to the mother-child relationship commonly found in mammals [6], with the hormone oxytocin playing an important role. Oxytocin secretion increases when an owner strokes or gazes at their dog [5,7,8,9], which might explain the health benefits of human–dog interactions. We sought to clarify the physiological mechanism by which interaction with pets stimulates oxytocin as measured in urine and saliva. Dog walking is a common activity for dog owners and their dogs [10,11]. Walking with a dog is almost certainly necessary for the health of the dog and should also be good for the health of the owner. However, many recent studies appear to be less positive [12,13,14], perhaps due to methodological concerns. In previous studies, hormones such as oxytocin and cortisol were the main indicators of the effects of walking [12], and heart rate variability analysis was the only neurological index [13,14]. Dog walking sometimes increases the owners’ oxytocin levels [12,13] and may reduce cortisol levels [15]. However, researchers have not established the mechanism by which human–dog interactions influence oxytocin and the autonomic nervous system. For example, the positive effects of interacting with dogs, such as walking and touching, remain uncertain, as some people experience elevated oxytocin and others do not. We re-examined the state of such hormones (Experiment I) along with the dynamics of nerves in the brain (Experiment II).

In this study we will evaluate the mechanism of action by focusing on the neurotransmitters involved in frailty (weakness of mind and body due to aging) and vitality in the context of dog walking (Experiment II). The noradrenergic nerve, which may be involved in discomfort, strongly influences frailty. Most of the noradrenaline in the brain is metabolized to MHPG (3-methoxy-4-hydroxyphenylethylene glycol) [16,17] by metabolic enzymes such as MAO (monoamine oxidase). Some MHPG is excreted in the blood and further diffused in saliva. Considering that MHPG metabolized by noradrenergic neural activity in the brain is transferred to saliva, it may be possible to assess noradrenergic neural activity in the brain by quantifying MHPG in saliva [17,18].

Dopamine secretion in the brain is involved in human vitality. Dopamine is converted to DOPAC (3,4-dihydroxyphenylacetic acid) by MAO, and DOPAC is converted to HVA (homovanillic acid) by COMT (catechol-*O*-methyltransferase) [19], which is abundant in the liver and kidneys. Similar to MHPG, therefore, DOPAC in saliva may reveal the activity of dopaminergic nerves in the brain. In addition, we measured GABA, which is an inhibitory neurotransmitter [20]. GABA in saliva probably originates in the brain.

## 2. Materials and Methods

### 2.1. Ethics Statement

The Ethics Committee at Yamazaki Animal Nursing University approved our experimental plans (approval numbers: 2021510-002 and 20210419-001) in accordance with the World Medical Association’s Declaration of Helsinki.

### 2.2. Participants

Thirty-four pairs of dogs and their owners participated in the study. The dogs included nine breeds, including mixes. Seventy-four percent of the owners were female, and they ranged in age from 45 to 75 years (Table 1). Both dogs and owners were healthy and not overweight.

### 2.3. Experiments I and II

We performed two experiments to investigate the effects of dog walking. For these experiments, owners walked with their dogs for about 2.0 km on courses that they regularly followed. Most owners had multiple walking courses. Basically, there was nothing that would hinder walking with a dog in any of the experimental walks. The dog–owner pairs walked their courses in 30 min. Saliva was collected during the walk, so the average speed was about 80 m per minute. Owners took two walks between 6 and 9 a.m. (morning) and between 5 and 8 p.m. (evening). The experimental walk was with a dog, and the control walk was without a dog. About half of both the walks with a dog and walks without a dog occurred in the morning and the other half occurred in the evening. In addition, each owner took both walks either in the morning or evening on separate days, with the dog walk occurring first. 

The dogs were allowed to sniff and they walked alongside their owners as usual. Participating dogs had some obedience training at universities and veterinary clinics, and there were no large differences in walking behaviour. However, saliva collection did differ between dogs that had sufficient training in brushing their teeth and dogs that did not. We told them to take a walk as usual, so there were some differences, but they did not seem to have a considerable impact on the experimental results.

We conducted the experimental walks from autumn to the spring of the following year, avoiding the hot summer (July to September).

In Experiment I, we assessed salivary oxytocin and cortisol as in previous studies [12,15,21]. In Experiment II, we assessed indicators of brain neural activity. As shown in Figure 1, participants collected their own saliva samples before the walk (Pre), 15 min into the walk (Walk-1), at the end of the 30 min walk (Walk-2), and 10 min after the end of the walk (Post). In the experimental walk, participants also collected saliva samples from their dogs.

We instructed the owners not to drink caffeine or use tobacco for least 120 min before saliva sampling [22]. Participants collected saliva samples by putting cotton swabs (Mentip^®^ hospital cotton swab 10P1508, JCB Industry Limited, Tokyo, Japan) into their mouths for about 1 min. They placed the collected saliva in a 10 mL separation tube, which they temporarily stored in an ice bath. A volunteer student took the ice bath, followed the owner and her/his dog inconspicuously, and received the saliva sample. Then, the student transferred the tube into a box containing dry ice (−45 °C), and carried it to a laboratory. The tubes were stored frozen at −80 °C in the laboratory until analysis.

Owners collected the dogs’ saliva samples in the same manner with the same kind of cotton swabs as they used for themselves. We instructed owners to brush their dogs’ teeth daily in preparation for the study to acclimate them to the owners sticking objects into their mouths. We also instructed participants to prevent their dogs from drinking water 30 min before sampling. Processing and storage procedures for the dog saliva samples were the same as for their owners’ samples. 

### 2.4. Measurements of Hormones

We thawed the frozen cotton swabs containing saliva in 10 mL separation tubes, centrifuged them at 3500 rpm for 15 min at 4 °C, and then used the separated fluids as samples. We measured hormone levels in saliva with the enzyme-linked immunosorbent assay (ELISA) kits, Enzo Biochem (Enzo Life Sciences Inc., Farmingdale, NY, USA), [23] and Oxford Biomedical Research (Rochester Hills, MI, USA) [24] for oxytocin and cortisol, respectively. These kits have reduced non-specific binding, and therefore are more accurate. The minimum sensitivities of cortisol and oxytocin were 5.0 and 15.6 pg/mL, respectively. We processed all samples in accordance with the kit manual protocols.

### 2.5. Measurements of Monoamines, Their Metabolites, and GABA

To deproteinize the saliva, we set 200 μL of saliva in a filter (CENTRICUT Ultramini W-MO-2, membrane pore diameter 0.2 μm, Kurabo Industries Co., Ltd., Osaka, Japan) and centrifuged it at 5000 G for 2 min at 4 °C. We then diluted the deproteinized filtrate 3- to 10-fold with 0.02 M acetic acid containing 10 μM EDTA 2Na and 10% methanol to measure monoamines (and their metabolites) and GABA, respectively.

We measured monoamines of noradrenaline (NA), adrenaline (AD), dopamine (DA), serotonin (5-HT), and their metabolites (including MHPG, and DOPAC) in saliva using high-performance liquid chromatography with an electrochemical detector (HPLC, EiCOM CO., LTD., Kyoto, Japan) and a 2.1φ × 150 mm column (EICOMPACK SC-5ODS) with a precolumn (PREPAK 3.0φ × 4.0 mm) (EiCOM). We set overvoltage to +750 mV against Ag/AgCl at 25 °C (HTEC-500, EiCOM). The mobile phase of the 01 M acetic acid–citrate buffer (pH 3.5) contained 0.013 mM EDTA-2Na, 190 mg/L sodium octane sulfonic acid, and 15% methanol pumped at a flow rate of 500 μL/min. Analysis duration was 30 min. We used the eDAQ PowerChrom (EiCOM, LTD.) software for determination.

Our procedures for measuring salivary GABA were similar, with a few differences. We added one volume (5 μL) of 4 mM o-phthaldialdehyde (OPA) and 2-mercaptoethanol solution (2-ME) (FUJIFILM Wako Chemicals Corporation, Osaka, Japan)/0.5 M carbonate buffer (pH 10.0) to 20 μL of saliva and allowed them to react for 2.5 min at room temperature. Next, we injected 10 μL of the sample into the HPLC/electrochemical detector with a graphite electrode set at +600 mV against Ag/AgCl (EiCOM) at 30 °C. Then, we separated GABA on a SA-50DS column (2.1φ × 150 mm) with a precolumn (PREPAK PC-04-CA 3.0φ × 4.0 mm) (EiCOM). The mobile phase consisted of 0.1 M phosphate buffer (pH 6.0), 0.13 mM EDTA 2Na, and 27% methanol pumped at a flow rate of 500 μL per min.

### 2.6. Statistics

We used Microsoft Office Excel (version 16.46) and Statcel 4 (2015 edition, OMS Publishing, Tokyo, Japan) for statistical analysis. For all outcome measures, we performed one-way ANOVAs and subsequent Tukey-Kramer and *t*-tests. We also computed the Pearson correlations between values of salivary monoamines, their metabolites, and GABA in owners and their dogs.

## 3. Results

Sixteen owner-dog pairs participated in Experiment I (from #1 to #16, Table 1), but only 10 pairs had complete saliva samples from both owners and dogs (four). In Experiment II, 14 of the 18 participating owner–dog pairs had complete samples (from #17 to #34, Table 1).

### 3.1. Effect on Dog Owners’ and Pet-Dogs’ Hormone Levels in Experiment I

The owners’ oxytocin levels increased during walks, both with and without dogs, after the pre-walk saliva samplings (Table 2), but the increases were not significant.

In addition, the owners’ saliva cortisol levels did not change on average over the course of either experiment or differ by walk condition (Table 2).

Oxytocin and cortisol levels in dogs were about four times higher and two times lower, respectively, than in their owners.

### 3.2. Effects on Owners’ Hormone Levels and Neural Activity (MHPG and GABA Levels) in Experiment II

The change in the hormonal levels of the dog owners in Experiment II (Table 3) was almost the same as in Experiment 1 (Table 2). That is, oxytocin tended to rise after the start of the walk. Cortisol was slightly higher on walks with dogs than without dogs, but none of the hormonal changes were significant. That is, oxytocin tended to rise after the start of the walk.

When owners walked alone, salivary MHPG gradually decreased, albeit not significantly (Figure 2). These MHPG values varied greatly among owners (range = 0.6 to 2116.6 ng/mL). As shown in Table 3, prior to dog walking, the owners’ MHPG levels were already low (mean = 79.4 ± 44.4 ng/mL) and then decreased during and after dog walking (Post mean = 28.4 ± 7.6 ng/mL). There was no significant difference between each order of walks, such as Walk-1(A) vs. Walk-1(D), whereas there was a significant difference between the groups without a dog and with a dog (*p* < 0.001, paired *t*-test). MHPG after dog walking was lower on average than after walking alone (*p* = 0.059, paired t-test) (Figure 2). The Pearson correlation coefficient between MHPG and oxytocin concentrations in saliva was 0.16 (*p* = 0.09). In other words, oxytocin showed a tendency to rise after the start of the walk.

The owners received a 39.6% increase in GABA when walking with their dogs (Table 3). The Pearson correlation coefficient between GABA and MHPG was −0.50 (*p* < 0.01) (Figure 3). However, only 20 out of 112 samples showed concentrations above 50 ng/mL of MHPG. 

Salivary concentrations of DOPAC ranged from 2.71 to 82.63 ng/mL (23.1 ± 3.1 ng/mL), with no apparent changes associated with walking.

### 3.3. Results for Pet Dogs in Experiment II

The dogs’ oxytocin levels did not significantly increase during and after walking relative to before walking (Table 4). However, the dogs’ oxytocin levels were about four times higher than their owners on average in Experiment II (*n* = 56 samplings for each group; dogs’ mean = 306.7 ± 10.2 pg/mL, owners’ mean = 65.2 ± 6.9 pg/mL; *p* < 0.001, paired *t*-test).

The pet dogs’ saliva cortisol levels did not change on average over the course of either experiment or differ by walk condition.

Owners’ MHPG levels were more than 3.6 times higher than those of their dogs on average (*n* = 42 (Walk1(D) + Walk2(D) + Post(D)) samplings; dogs’ mean = 7.6 ± 3.0 ng/mL, owners’ mean = 27.9 ± 4.2 ng/mL).

Salivary DOPAC was 0.00 to 35.05 ng/mL (4.3 ± 1.1 ng/mL) in pet dogs, which was significantly lower than the levels of their owners (*p* < 0.0001, *t*-test), although there were no changes with walking.

The dogs’ GABA levels (mean = 3.1 ± 0.5 μM, *n* = 20) correlated negatively with their MHPG levels (*r* = −0.47, *p* < 0.05).

## 4. Discussion 

### 4.1. Dog Owners

The effects of dog walking on dog owners were consistent with prior research [12,13]. Walking may have a positive effect of elevated oxytocin on some people without dogs, but not on others. To date, there is no significant difference in other parameters, such as heart rate variability analysis [12]. From the results of salivary oxytocin and cortisol in Experiment I, we concluded that dog walking had a good effect on some people, as in one review [25]. 

The large standard errors highlight the substantial individual differences in MHPG (the major metabolite of norepinephrine) levels. Across all four samplings in Experiment II, the overall MHPG means for dog walking (41.7 ± 12.8, *n* = 56) and walking alone (279.6 ± 71.3 ng/mL, *n* = 56) were clearly significantly different (Table 3 and Figure 2). Thus, dog walking clearly reduced MHPG in saliva, which may represent decreased stress [26,27].

In some previous studies [28,29], salivary MHPG values were below 40 ng/mL. In this study, 20 out of 112 samples showed concentrations above 50 ng/mL of MHPG. In one previous study [28], salivary MHPG was measured using gas chromatography coupled with mass spectrometry (Hitachi-M80B, Hitachi, Japan), while, in another study [29], it was assessed by the same HPLC system as in our study. Unlike previous studies, the subjects, who showed more than 50 ng/mL of MHPG in this study, were all 65 years or older. Although they were physically and mentally healthy, they may be at risk of frailty. The level of MHPG in saliva indicates the activity of noradrenergic nerves in the brain. It is also a good indicator of stress [26,30] but is not yet considered to be a condition of frailty. In aging, the decline in physiological reserves increases the vulnerability to stress, and agility and mobility decrease due to loss of muscle strength. In geriatrics, frailty has received increasing attention with respect to pathophysiology, diagnosis, and prevention of long-term care [31,32]. Therefore, it is necessary to make medical professionals and the public aware of frailty, as they can promote the prevention of long-term care and reduce the number of elderly people requiring long-term care. Despite common notions that frailty is irreversible and increases with age, frailty might be reversed with proper intervention. We think this point is important, especially given that there is no research on the matter.

Exposure to a stress task generally activates the hypothalamic–pituitary–adrenal (HPA) axis, resulting in the secretion of cortisol from the adrenal cortex. We also measured the concentration of salivary cortisol, whose secretion can be studied in two main ways, examinations of acute stress reactivity and of the basal circadian patterns. It would not be appropriate to compare changes in cortisol and MHPG, because the walk in this study took place in the morning or the evening.

The relationship between MHPG and GABA (Table 3 and Figure 3) is consistent with GABA suppressing noradrenergic nerves. Although further research is needed, walking with a dog can be enjoyable and might therefore stimulate GABA secretion in the brain [33,34]. DOPAC, one of the metabolites to evaluate the activity of dopaminergic nerves in the brain, was measured in the saliva of owners and pet dogs but DOPAC was not significantly correlated with oxytocin or GABA (*p* > 0.05).

### 4.2. Pet Dogs

The dogs’ oxytocin levels were about four times higher than those of their owners on average, although the oxytocin levels did not significantly increase during walking (Table 2 and Table 4). The dogs’ mean (306.7 ± 10.2 pg/mL) was almost the same as the value in previous studies [35,36]. The dogs’ higher levels of oxytocin may be due to their affection for humans [9].

The MHPG in the dogs’ saliva was about one-quarter and the cortisol levels were half those of humans (owners), suggesting that the dogs’ stress levels were clearly low. The dogs’ GABA levels correlated negatively with their MHPG levels, mirroring a similar relationship in owners.

In both the owners and the pet dogs, the assays of saliva samples were performed in the order of oxytocin, cortisol, monoamine (including metabolites), and GABA. If the sample amount was not sufficient (400 μL or less), the measurement had to be omitted in the reverse order. Therefore, the samples of the dogs’ measuring GABA were only 20. 

### 4.3. A Relationship between a Pet Dog and the Owner

How does a pet dog affect its owner’s health? The dog-less walk literally means that a dog was not on the side of the owner. Walking with the dog significantly reduced salivary MHPG, which might have occurred via GABAergic nerves, and apparently alleviated the frailty condition. This may be the “characteristic” of the dog brought about by domestication [9]. Activities like dog walking can be enjoyable (setting off the GABAergic system) for both pet dogs and their owners.

## 5. Conclusions

The change in the owners’ oxytocin levels when walking was not due to the presence or absence of dogs. However, walking with a dog clearly suppressed the activity of the noradrenergic nerve. Such a positive effect may lessen the frailty of elderly persons who walk their dogs. It seems that GABAergic nerves, one kind of inhibitory neuron, may suppress activity of noradrenergic nerves.

## 6. Limitations and Speculation

The small sample sizes are a limitation to our experiments. Insufficient saliva samples further decreased effective sample sizes. From some dogs, it is challenging to collect enough saliva for testing (200 μL for oxytocin, 100 μL for cortisol, and 100 μL each for monoamine metabolites and GABA, for a total of 400 μL). It was not easy to collect 400 μL of saliva in small dogs. After this study, we tried many times to collect saliva from small dogs, but we failed to collect 400 μL even once. Although it may be possible to collect such amounts from dogs weighing 10 kg or more, there are few such medium and large dogs in Japan today.

Dogs and humans have had a close relationship for more than 10,000 years. Dogs had oxytocin levels several times higher than those of humans. Dogs also had very low activity in their noradrenergic nerves but had high GABA activity. These results may suggest that it is beneficial for older people to live with dogs.

## Figures and Tables

**Figure 1 animals-11-02732-f001:**
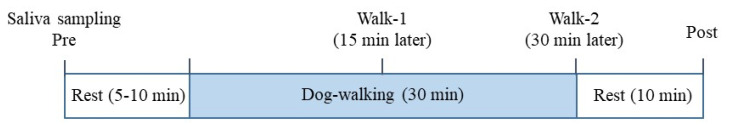
Schedule of four saliva samplings (Pre, Walk-1, Walk-2, and Post).

**Figure 2 animals-11-02732-f002:**
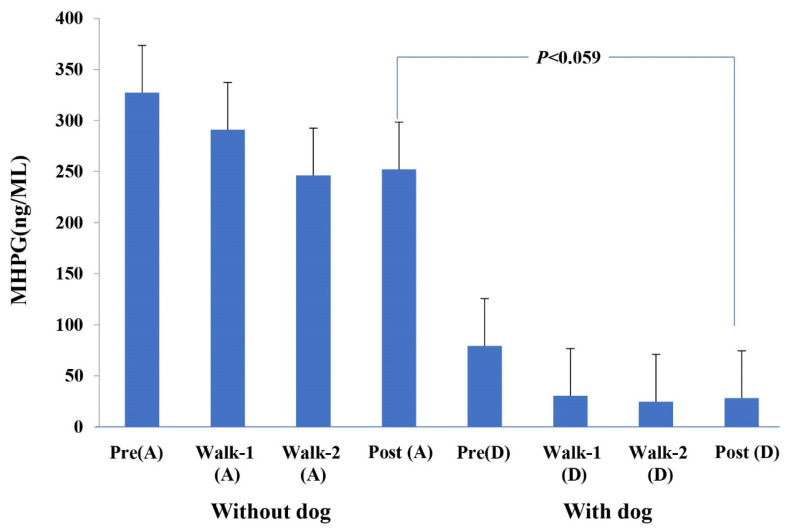
MHPG levels in owners’ saliva, Experiment II.

**Figure 3 animals-11-02732-f003:**
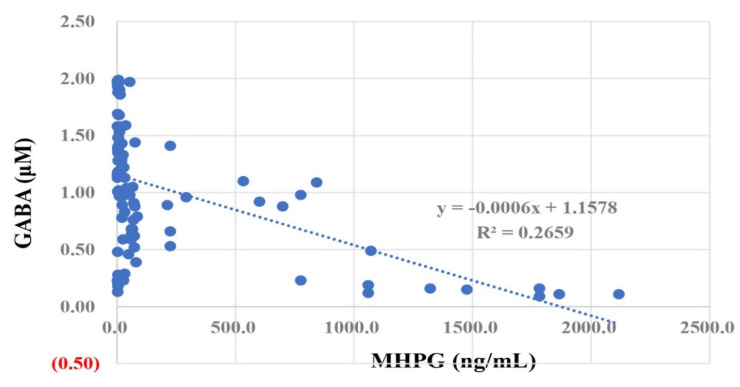
Scatter plot of salivary MHPG (horizontal axis) and GABA (vertical axis).

**Table 1 animals-11-02732-t001:** List of participants in the experiments.

	Owner		Dog			Owner		Dog	
	Sex	Age	Breeds	Age		Sex	Age	Breeds	Age
1 *	M	71	Labrador Retriever	11	18 *	F	66	Miniature Dachshund	4
2	F	65	Golden Retriever	2	19	F	70	Golden Retriever	2
3	M	71	Shiba	12	20	M	55	White Shepherd	4
4	F	58	Golden Retriever	10	21	F	50	White Shepherd	4
5	F	59	Shiba	7	22	F	45	Shiba	5
6	M	75	Golden Retriever	3	23	M	50	Shiba	5
7 *	F	70	Miniature Dachshund	5	24	M	50	White Shepherd	4
8 *	F	66	Mix	10	25	F	50	Toy Poodle	7
9	M	63	Labrador Retriever	3	26 *	F	50	Miniature Dachshund	7
10	F	61	Labrador Retriever	12	27	F	50	Mix	5
11	F	60	Labrador Retriever	9	28	M	55	Mix	5
12	F	65	Golden Retriever	1	29	F	65	Labrador Retriever	8
13 *	F	60	Toy Poodle	2	30	F	67	Labrador Retriever	9
14 *	F	53	Miniature Dachshund	2	31	F	68	Miniature Schnauzer	7
15	F	65	Shiba	1	32	F	50	Shiba	5
16 *	F	60	Toy Poodle	8	33 *	F	53	Toy Poodle	7
17	F	65	Shiba	7	34 *	M	75	Miniature Schnauzer	2

* Samples from participants 1, 7, 8, 13, 14, 16, 18, 26, 33, and 34 removed from data analysis.

**Table 2 animals-11-02732-t002:** The concentrations of oxytocin and cortisol in dog owners and pet dogs in Experiment I.

			Oxytocin		Cortisol	
	Stages	N	Means	S.E.	Means	S.E.
Walk without dog	Pre(A)	10	59.24	23.54	0.38	0.06
	Walk1(A)	10	82.38	20.13	0.33	0.05
	Walk2(A)	10	92.44	27.68	0.28	0.04
	Post(A)	10	92.27	34.20	0.25	0.03
Walk with dog	Pre(D)	10	57.96	11.45	0.50	0.11
	Walk1(D)	10	117.11	28.54	0.38	0.08
	Walk2(D)	10	85.99	20.10	0.41	0.10
	Post(D)	10	77.22	17.88	0.34	0.06
Dog	Pre(D)	10	275.0	25.1	0.16	0.01
	Walk1(D)	10	329.3	26.9	0.20	0.02
	Walk2(D)	10	312.9	21.8	0.20	0.03
	Post(D)	10	325.3	27.8	0.17	0.02

**Table 3 animals-11-02732-t003:** The concentrations of oxytocin, cortisol, MHPG, and GABA during walk alone (**A**) or walk with dog (**B**) in Experiment II.

**A. Walking without Dog**					
	**Substances**	**Units**	**N**	**Means**	**S.E.**
Pre(A)	Oxytocin	pg/mL	14	62.30	9.14
	Cortisol	μg/dL	14	0.38	0.06
	MHPG	ng/mL	14	264.10	134.31
	GABA	μM	14	1.15	0.13
Walk1(A)	Oxytocin	pg/mL	14	93.40	19.47
	Cortisol	μg/dL	14	0.33	0.05
	MHPG	ng/mL	14	274.59	142.76
	GABA	μM	14	0.98	0.10
Walk2(A)	Oxytocin	pg/mL	14	83.18	16.56
	Cortisol	μg/dL	14	0.28	0.04
	MHPG	ng/mL	14	294.59	162.75
	GABA	μM	14	0.84	0.14
Post(A)	Oxytocin	pg/mL	14	60.96	15.39
	Cortisol	μg/dL	14	0.25	0.03
	MHPG	ng/mL	14	285.11	144.83
	GABA	μM	14	0.62	0.12
**B. Walking with Dog**					
	**Substances**	**Units**	**N**	**Means**	**S.E.**
Pre(D)	Oxytocin	pg/mL	14	60.96	14.23
	Cortisol	μg/dL	14	0.50	0.11
	MHPG	ng/mL	14	79.36	44.37
	GABA	μM	14	0.91	0.13
Walk1(D)	Oxytocin	pg/mL	14	64.39	12.41
	Cortisol	μg/dL	14	0.38	0.08
	MHPG	ng/mL	14	30.55	8.12
	GABA	μM	14	1.21	0.10
Walk2(D)	Oxytocin	pg/mL	14	68.35	12.50
	Cortisol	μg/dL	14	0.41	0.10
	MHPG	ng/mL	14	28.51	6.70
	GABA	μM	14	1.36	0.13
Post(D)	Oxytocin	pg/mL	14	67.06	15.39
	Cortisol	μg/dL	14	0.34	0.06
	MHPG	ng/mL	14	28.35	7.63
	GABA	μM	14	1.27	0.14

**Table 4 animals-11-02732-t004:** Oxytocin and cortisol concentrations in dog’s saliva, Experiment II.

		Oxytocin			Cortisol	
	Stages	N	Means	S.E.	Means	S.E.
Dog	Pre(D)	14	284.5	19.0	0.17	0.03
	Walk1(D)	14	332.7	20.9	0.16	0.02
	Walk2(D)	14	345.0	25.2	0.14	0.01
	Post(D)	14	312.2	28.9	0.21	0.05

## Data Availability

Not applicable.

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
