# Peer review of "Hormonal and Neurological Aspects of Dog Walking for Dog Owners and Pet Dogs"

_animals, 2021, doi:10.3390/ani11092732_

Round 1

Reviewer 1 Report

Overall this is a well-executed and presented study and I commend the authors on their good work. What has arisen in the study is the need for further work and I would like to see the authors highlight this. As well, we need some more background on any variables for the dog walking scenarios.

There are a few parts of the experiment where I would like to see more information.

For the 30 minutes walks that the owner and dog were habituated too: What was around the walkers? Did it impact? Were there other dogs, people, bicycles, cars, interruptions that could have impacted the effect on both the dog and the person? Impacted differently on different pairs?

Pace of walk? It would have been good to have had a clearer overview of what walks were selected and why. Did the walkers run at differing tempos at any point? Were the dogs allowed to sniff the entire time or required to walk alongside their owners? How were these walks conducted? Did it vary between the dog-owner pairs?

Fitness levels: were all dogs and owners used to 30 minute walks each day? was this the usual? or did some do 30, others 2 X 30, others 1 hour usually? Were others usually running? others only sniffing? Others a mix? Was this their usual pace and did it differ from the other pairs?

In addition, there is a complication with the smaller dogs. Perhaps, this could have been discovered in a small pilot test and different larger dogs added to this test. But either way, please do be sure to add information on the kinds of walks, pace, fitness levels and any differences between the pairs in the experimental setup section. And also add to the Future Work section, with a selection of larger dogs, and ensure that any other variables are included in any analyses. 

Regardless of these provisions, I am happy to support the paper and look forward to seeing further work in this area.

Author Response

Response to Reviewer 1 Comments

Point 1: There are a few parts of the experiment where I would like to see more information.

For the 30 minutes walks that the owner and dog were habituated too: What was around the walkers? Did it impact? Were there other dogs, people, bicycles, cars, interruptions that could have impacted the effect on both the dog and the person? Impacted differently on different pairs?

Pace of walk? It would have been good to have had a clearer overview of what walks were selected and why. Did the walkers run at differing tempos at any point? Were the dogs allowed to sniff the entire time or required to walk alongside their owners? How were these walks conducted? Did it vary between the dog-owner pairs?

Fitness levels: were all dogs and owners used to 30 minute walks each day? was this the usual? or did some do 30, others 2 X 30, others 1 hour usually? Were others usually running? others only sniffing? Others a mix? Was this their usual pace and did it differ from the other pairs?

Response 1: We conducted the experiment with individual dog-owner pairs on walking courses they used regularly. Most owners had multiple walking courses. A course of about 2.0 km was used for the experiments. Basically, there was nothing that would hinder walking with a dog in any of the experimental walks.

The dog-owner pairs walked their courses in 30 minutes. Saliva was collected during the walk, so the average speed was about 80 meters per minute. Otherwise, the walks occurred just like their usual walks, and the variation across pairs in walking behavior paralleled their natural, pre-existing variation. This variation did not seem related to the experimental results. We conducted the experimental walks from autumn to the spring of the following year, avoiding the hot summer (July to September).

Yes, the dogs were allowed to sniff, and they walked alongside their owners as usual. Participating dogs had some obedience training at universities and veterinary clinics, and there were no large differences in walking behavior. However, saliva collection did differ between dogs that had sufficient training in brushing their teeth and dogs that did not. We told them to take a walk as usual, so there were some differences, but it didn't seem to have a big impact on the experimental results.

We have added relevant sentences on these points in the Materials and Methods section.

Point 2: In addition, there is a complication with the smaller dogs. Perhaps, this could have been discovered in a small pilot test and different larger dogs added to this test. But either way, please do be sure to add information on the kinds of walks, pace, fitness levels and any differences between the pairs in the experimental setup section. And also add to the Future Work section, with a selection of larger dogs, and ensure that any other variables are included in any analyses.

Regardless of these provisions, I am happy to support the paper and look forward to seeing further work in this area.

Response 2: We have revised the Materials and Methods section to address these points. Future studies must clarify: 1) Why do dogs have oxytocin levels several times higher than humans? 2) Why does human oxytocin rise when interacting with dogs?, and 3) What is the trigger? In some previous studies, we found that an owner’s oxytocin increases when a dog looks at the owner’s face. At that time, we came up with the important idea to investigate the neural activities in the brain. HPLC can be now used to measure monoamines, their metabolites, and GABA. We would like to evaluate the hypothesis that the rise in oxytocin is triggered by the "pleasure" of animals and humans.

We have also revised the Discussion to address this reviewer’s points. Our ultimate goal is to assess whether living with a dog promotes the health of the elderly. We hope that many researchers will undertake this work.

We are deeply grateful for this reviewer’s important suggestions.

Reviewer 2 Report

COMMENTS TO AUTHORS

The aim of the present study was to investigate the effect of the activity of dog walking on a range of physiological and neural markers of dogs and owners. In two experiments, saliva samples were collected from both dogs and owners at different time-points (before, during and after the walk) in two conditions (owner walking alone and owner walking with dog). Results showed that the activity of walking a dog, as compared to walking alone, lead to a reduction in MHPG in dog owners, but did not affect cortisol or oxytocin levels. GABA and MHPG were also found to correlate negatively in both dogs and owners.

I think the current manuscript has some data that may be worth being published, but the paper needs substantial revising. In what follows I detail my comments and concerns.

First, the research question needs to be clarified and the rationale of the study needs to be strengthened. The authors measured several different physiological and neural markers, but the reason for choosing these markers and the connection between them is not presented in a clear form and/or is not sufficiently substantiated by previous literature. I would start addressing this by asking the authors to clearly state what did they actually aim at addressing with this study -- was it the well-being of the dogs and owners? Please clarify how the different markers that were measured relate to each other and to the research question (that needs to be clarified on its own!). In its current form, everything seems a bit disconnected. Revising the Introduction section is mandatory.

Secondly, the presentation of Materials and Methods and Results is confusing and difficult to follow. The paper will definitely benefit from a better organization and needs further clarification. More specifically:

Materials and Methods

The Participants sub-section should already include all the information on the sample size – that the authors started started with 34 dog-owner pairs but that the final sample size was 24 (i.e., the information that is only latter presented in lines 145-148).

Moreover, since the authors divided the study in two experiments, these should be presented and explained (as well as their corresponding sample sizes) right in the beginning of the Materials and Methods section (maybe immediately after Participants).

Lines 80-83: The methodology is not clearly explained. Did each owner have four walks, two alone and two with a dog? Were they all in the same day? What was the order of the walks? – which ones were with and without dog? Was this the case for each experiment, i.e., were there four walks per experiment? All this information needs to be clearly stated.

Additionally, at this point, it is not clear if the same saliva samples were used for all the performed measurements. It only gets clear on line 242 (Discussion!) that the same samples were used for measuring all the markers and that a given order was chosen.

Section 2.5. -- So far the authors only referred cortisol, oxytocin, MHPG and GABA (and monoamines in the Simple Summary) as variables under study. At this point, other markers are presented (e.g., DOPAC). Please ensure you refer to the measured variables consistently across the entire manuscript.

Section 2.6. -- Did the authors test for normality of the data? (since parametric tests were chosen)

Thirdly, the presentation of results is confusing. I would suggest that the authors re-organize this section and report/present results by experiment -- first the results for Experiment I and then for Experiment II (first for owners and then for dogs’ data within each experiment, for example). Moreover, I suggest that the authors re-think the presentation of results in tables and graphs. I would only keep Figure 3 (where we can see the most interesting results) and drop all the remaining graphs, reporting all the remaining results in tables. It is also important that all results are reported either in text, tables or graphs in the Results section -- Some data are missing, as for example the results “without dog” in Table 3, and some other results are only presented for the first time in the Discussion section (e.g., lines 229-232).

Among other things, until we get to this section, the authors state that Experiment I aimed at measuring oxytocin, but when we get to the Results section oxytocin results are reported for both experiments.

Figure 3 – The results of this figure are intriguing and they should be addressed in the Discussion. What explains the difference between Pre(a) and Pre(d)? Does it have to do with the order of the walks, which is not currently reported in the paper?

Section 3.4. -- Why compare dogs’ with owners’ levels? Does this type of interspecific comparisons make sense biologically? (same for Discussion). Be it as it may, this was not pointed out before as a hypothesis the authors had. I would only report the results of dogs across the different time-points of the walks.

Lines 186-188: “pet dogs’ saliva cortisol levels did not change on average over the course of either experiment or differ by walk condition” – Were there two walk conditions for the dogs? Nothing was referred about this so far.

Line 165-166: What is the reason/rationale for performing this correlation (MHPG and oxytocin)? This was not a correlation thought in advance.

What about reporting the results for MHPG across time-points?

Finally, please make sure all the statistical results are reported here (and remove the reported stats in Discussion). Importantly, do not refer at differences if they were not significant!

At last, some points on the Discussion section that need correction/consideration:

Lines 204-206: “From Experiment I, the results of salivary oxytocin and cortisol, we concluded that dog walking had a good effect on some people, as in one 205 review [23].” – How is this? There were no (statistically significant) differences.

Lines 214-200: Only now the authors clarify and address what levels of MGPH > 50 ng/ml mean (see lines 172-173, where the reader does not have any background to understand what the authors are talking about).

Lines 229-232 – Where is this result reported in the Results section?

Subsection 4.2. -- See my concern on interspecific comparisons above. Additionally, the last paragraph belongs to Materials and Methods.

Line 254-255: “The change in owners’ oxytocin levels when walking was not due to the presence or absence of dogs.” – Which difference? No difference was found.

Lines 258-260: “Further research may clarify how interaction with dogs of the owner involved with this process that arose during tens of thousands of years of dogs’ companionship with humans.” – Please clarify/rephrase this sentence; it is not intelligible.

Minor suggestions and typos:

Lines 35-37: I suggest changing “healing” to “beneficial”

Line 170: The use of the word “finally” does not seem adequate here

Line 231: One should read “…but those did not significantly correlate…”

Line 246: “dog” instead of “dogs”

Line 251: remove “a” after “Activities like…”

Author Response

Response to Reviewer 2 Comments

Point 1: First, the research question needs to be clarified and the rationale of the study needs to be strengthened. The authors measured several different physiological and neural markers, but the reason for choosing these markers and the connection between them is not presented in a clear form and/or is not sufficiently substantiated by previous literature. I would start addressing this by asking the authors to clearly state what did they actually aim at addressing with this study -- was it the well-being of the dogs and owners? Please clarify how the different markers that were measured relate to each other and to the research question (that needs to be clarified on its own!). In its current form, everything seems a bit disconnected. Revising the Introduction section is mandatory.

Response 1: As we stated in the abstract, walking with a dog is almost certainly necessary for the health of the dog and would also be good for the health of the owner. However, many recent studies appear to be less positive, perhaps due to methodological problems. In previous studies, hormones such as oxytocin and cortisol were the main indicators of the effect of walking, and heart rate variability analysis was the only neurological index. We wanted to re-examine the state of such hormones along with the dynamics of nerves in the brain, so we conducted this research. Fortunately, we could use saliva samples to do both. We have revised the Introduction to state this more clearly.

Point 2: The Participants sub-section should already include all the information on the sample size – that the authors started with 34 dog-owner pairs but that the final sample size was 24 (i.e., the information that is only latter presented in lines 145-148).

Moreover, since the authors divided the study in two experiments, these should be presented and explained (as well as their corresponding sample sizes) right in the beginning of the Materials and Methods section (maybe immediately after Participants).

Response 2: We revised the Materials and Methods section to address these points.

Point 3: Lines 80-83: The methodology is not clearly explained. Did each owner have four walks, two alone and two with a dog? Were they all in the same day? What was the order of the walks? – which ones were with and without dog? Was this the case for each experiment, i.e., were there four walks per experiment? All this information needs to be clearly stated.

Additionally, at this point, it is not clear if the same saliva samples were used for all the performed measurements. It only gets clear on line 242 (Discussion!) that the same samples were used for measuring all the markers and that a given order was chosen.

Response 3: Most owners took walks with their dogs twice a day (morning and evening). About half of both the walks with a dog and walks without a dog occurred in the morning and the other half occurred in the evening. Also, each owner took both walks either in the morning or evening on separate days, with the dog walk occurring first. In other words, no owner took a walk with a dog in the morning but a walk without a dog in the evening, or vice versa. We sampled saliva 4 times on a walk day (pre, walk 1, walk 2, & post within 1 hour). We added text in the Materials and Methods section to address these points.

Point 4: Additionally, at this point, it is not clear if the same saliva samples were used for all the performed measurements. It only gets clear on line 242 (Discussion!) that the same samples were used for measuring all the markers and that a given order was chosen.

Section 2.5. -- So far the authors only referred cortisol, oxytocin, MHPG and GABA (and monoamines in the Simple Summary) as variables under study. At this point, other markers are presented (e.g., DOPAC). Please ensure you refer to the measured variables consistently across the entire manuscript.

Section 2.6. -- Did the authors test for normality of the data? (since parametric tests were chosen)

Response 4: The same saliva sample was used for measuring all the markers (the two hormones, monoamines, their metabolites, and GABA,). First, oxytocin was measured, then cortisol. The remaining saliva was used to measure monoamines, their metabolites, and GABA.

Substances released from the nerves of the brain into the blood then diffuse into saliva. We avoided serotonin and its metabolites, which are abundant in the periphery, even though they are produced by nerves in the brain.

Yes, we performed parametric tests.

Point 5: The presentation of results is confusing. I would suggest that the authors re-organize this section and report/present results by experiment -- first the results for Experiment I and then for Experiment II (first for owners and then for dogs’ data within each experiment, for example). Moreover, I suggest that the authors re-think the presentation of results in tables and graphs. I would only keep Figure 3 (where we can see the most interesting results) and drop all the remaining graphs, reporting all the remaining results in tables. It is also important that all results are reported either in text, tables or graphs in the Results section -- Some data are missing, as for example the results “without dog” in Table 3, and some other results are only presented for the first time in the Discussion section (e.g., lines 229-232).

Among other things, until we get to this section, the authors state that Experiment I aimed at measuring oxytocin, but when we get to the Results section oxytocin results are reported for both experiments.

Response 5: Following the reviewer's suggestion, we have divided the Results into Owners and Dogs sections, within the description of each experiment.

We have left Figures 3 and 4 as they are and changed Figures 2 and 5 to tables. Table 3 shows the data of the "walking without a dog" experiment. As the reviewer commented, Figure 3 is the most important of this paper. Next, Figure 4, which shows the relationship between MHPG and GABA, is the next most important one. Table 3 is not even necessary compared to Figures 3 and 4. Few authors have reported the average values of oxytocin, monoamine metabolites, and even GABA in their research, therefore, we believe it is important to include this information.

Point 6: Figure 3 – The results of this figure are intriguing and they should be addressed in the Discussion. What explains the difference between Pre(a) and Pre(d)? Does it have to do with the order of the walks, which is not currently reported in the paper?

Section 3.4. -- Why compare dogs’ with owners’ levels? Does this type of interspecific comparisons make sense biologically? (same for Discussion). Be it as it may, this was not pointed out before as a hypothesis the authors had. I would only report the results of dogs across the different time-points of the walks.

Lines 186-188: “pet dogs’ saliva cortisol levels did not change on average over the course of either experiment or differ by walk condition” – Were there two walk conditions for the dogs? Nothing was referred about this so far.

Line 165-166: What is the reason/rationale for performing this correlation (MHPG and oxytocin)? This was not a correlation thought in advance.

What about reporting the results for MHPG across time-points?

Finally, please make sure all the statistical results are reported here (and remove the reported stats in Discussion). Importantly, do not refer at differences if they were not significant!

Response 6: There was no significant difference between each order of walks, like Walk-1(A) vs Walk-1(D), whereas there was a significant difference between the groups without dog and with dog (P < 0.001). We have added these in Results section.

The level of MHPG in saliva indicates the activity of noradrenergic nerves in the brain. It's also a good indicator of stress, but is not yet considered to be a condition of frailty. In aging, the decline in physiological reserves increases the vulnerability to stress, and agility and mobility decrease due to loss of muscle strength. In geriatrics, frailty has received increasing attention with respect to pathophysiology, diagnosis, and prevention of long-term care. Therefore, it is necessary to make medical professionals and the public aware of frailty, as they can promote the prevention of long-term care and reduce the number of elderly people requiring long-term care. Despite common notions that frailty is irreversible and increases with age, frailty might be reversed with proper intervention. We think this point is important, especially given that there is no research on the matter. Accordingly, we believe the Pre-Post comparison is essential. We have modified the part of the Discussion related to dogs’ saliva cortisol, the correlation between MHPG and oxytocin, and statistical results.

Point 7: Lines 204-206: “From Experiment I, the results of salivary oxytocin and cortisol, we concluded that dog walking had a good effect on some people, as in one 205 review [23].” – How is this? There were no (statistically significant) differences.

Lines 214-200: Only now the authors clarify and address what levels of MGPH > 50 ng/ml mean (see lines 172-173, where the reader does not have any background to understand what the authors are talking about).

Lines 229-232 – Where is this result reported in the Results section?

Subsection 4.2. -- See my concern on interspecific comparisons above. Additionally, the last paragraph belongs to Materials and Methods.

Line 254-255: “The change in owners’ oxytocin levels when walking was not due to the presence or absence of dogs.” – Which difference? No difference was found.

Lines 258-260: “Further research may clarify how interaction with dogs of the owner involved with this process that arose during tens of thousands of years of dogs’ companionship with humans.” – Please clarify/rephrase this sentence; it is not intelligible.

Minor suggestions and typos:

Lines 35-37: I suggest changing “healing” to “beneficial”

Line 170: The use of the word “finally” does not seem adequate here

Line 231: One should read “…but those did not significantly correlate…”

Line 246: “dog” instead of “dogs”

Line 251: remove “a” after “Activities like…”

Response 7: In both Experiment I and Experiment II, oxytocin increased in some owners while walking with their dogs, while for other owner’s oxytocin decreased or did not change. There was no significant difference when averaging across owners, but for some individuals, walking has shown good results.

Participants in this study included owners who showed high values of MHPG that had not been reported in previous papers. In other words, we wanted to show that it was clearly different from the previous values and emphasize that it might indicate a frailty state.

We have deleted the sentences “Further research may clarify how interaction with dogs of the owner involved with this process that arose during tens of thousands of years of dogs’ companionship with humans.”

Lines 35-37: I suggest changing “healing” to “beneficial”; We have changes.

Line 170: The use of the word “finally” does not seem adequate here; We have deleted.

Line 231: One should read “…but those did not significantly correlate…”; We have modified.

Line 246: “dog” instead of “dogs”; We have modified.

Line 251: remove “a” after “Activities like…”; We have removed.

We have thoroughly revised the Discussion in response to the various comments from this reviewer.

We are deeply grateful to all the reviewers for their important suggestions.